# Immunotherapies and Metastatic Cancers: Understanding Utility and Predictivity of Human Immune Cell Engrafted Mice in Preclinical Drug Development

**DOI:** 10.3390/cancers12061615

**Published:** 2020-06-18

**Authors:** Tiina E. Kähkönen, Jussi M. Halleen, Jenni Bernoulli

**Affiliations:** 1OncoBone Ltd., Kalimenojankuja 3 C 4, FI-90810 Kiviniemi, Finland; jussi.halleen@oncobone.com; 2Institute of Biomedicine, University of Turku, Kiinamyllynkatu 10, FI-20520 Turku, Finland; jennibernoulli@gmail.com

**Keywords:** humanized mice, human immune system, preclinical oncology model, metastasis model, immunotherapy, efficacy, safety

## Abstract

Metastases cause high mortality in several cancers and immunotherapies are expected to be effective in the prevention and treatment of metastatic disease. However, only a minority of patients benefit from immunotherapies. This creates a need for novel therapies that are efficacious regardless of the cancer types and metastatic environments they are growing in. Preclinical immuno-oncology models for studying metastases have long been limited to syngeneic or carcinogenesis-inducible models that have murine cancer and immune cells. However, the translational power of these models has been questioned. Interactions between tumor and immune cells are often species-specific and regulated by different cytokines in mice and humans. For increased translational power, mice engrafted with functional parts of human immune system have been developed. These humanized mice are utilized to advance understanding the role of immune cells in the metastatic process, but increasingly also to study the efficacy and safety of novel immunotherapies. From these aspects, this review will discuss the role of immune cells in the metastatic process and the utility of humanized mouse models in immuno-oncology research for metastatic cancers, covering several models from the perspective of efficacy and safety of immunotherapies.

## 1. Immune Cells and Metastasis

Primary, localized cancer can be sufficiently treated, resulting in survival rates above 90% for most cancers. The majority of cancer-related deaths are caused by metastases that remain mostly incurable [1,2,3]. To enable metastasis formation, tumor cells need to go through the metastatic process where they change their properties, allowing them to leave the primary tumor, enter the vasculature, survive in the circulation, extravasate, home to new tissue microenvironment, and lastly, form a secondary tumor, which in many cases becomes resistant to many standard-of-care therapies [4].

The metastasis process is influenced by many cells in the local tumor microenvironment (TME) including different immune cells. Blomberg and colleagues have reviewed the parts of the metastatic processes that can be regulated by immune cells [5]. These include 1) tumor promoting inflammation; 2) regulation of cancer cell invasion and intravasation; 3) regulation of circulating tumor cell survival and extravasation; and 4) effects on pre-metastatic niche. The next paragraphs provide a short overview about the influence of different immune cells during the metastatic process.

In a tumor promoting inflammation, immune cells create an immunosuppressive microenvironment that enhances tumor growth and progression [6]. A tumor promoting inflammation is driven by tumor-educated myeloid cells such as tumor-associated macrophages (TAMs) and tumor-associated neutrophils (TANs) that can produce immunosuppressive cytokines (such as arginase 1, monocyte chemoattractant protein 1, interleukin (IL) 6 and 8, and T cell inhibitory molecules (such as programmed cell death 1 (PD-1) and programmed death ligand 1 (PD-L1)) [7,8,9]. TANs can further promote metastasis by suppressing and inducing apoptosis of cytotoxic CD8+ T cells [10] and increasing CD4+ T cells, for example through metabolite secretion [11]. The immunosuppressive microenvironment is enhanced by regulatory T cells (Tregs) that also express high levels of immune checkpoint molecules such as PD-1. Patients with prostate cancer bone metastasis have high levels of functional Tregs in the bone marrow, causing the immunosuppressive TME [12]. Regulatory B cells (Bregs) can promote immunosuppressive microenvironment even further by converting CD4+ T cells into Tregs [13,14] and inducing myeloid cells to become immunosuppressive by increasing expression of PD-L1 and production of immunosuppressive cytokines such as transforming growth factor β (TGFβ) and IL-10 [15]. Furthermore, tumor-associated regulatory B cells are needed for TGFβ-dependent pro-metastatic function of myeloid-derived suppressor cells (MDSCs) [16]. Furthermore, dendritic cells (DCs) are suppressed in the TME, leading to low antigen presentation and lowered immune responses [17].

Immune cells can modulate extracellular matrix (ECM) composition and blood and lymphoid vessel formation, and increase motility of cancer cells, leading to increased intravasation and possibility for formation of metastases. ECM can be modulated by TAMs [18,19] and TANs [20]. ECM can be re-modelled by the secretion of matrix metalloproteinases (MMPs), many of which also have direct effects on immune responses and immune suppression [21,22,23,24]. TAMs, TANs and mast cells can increase blood vessel formation and the number and density of vessels in tumors also by modulating ECM [25,26,27,28].

A crucial factor for tumor progression in tissue microenvironment is hypoxia that increases the formation of metastases, and distinct metastatic sites are differentially hypoxic. For example, one of the most hypoxic environments is bone marrow, where hypoxia affects not only tumor cells but also bone marrow and immune cells [29,30]. Hypoxic tissue microenvironment can increase the presence of MDSCs, leading to immune escape and decreased M1 macrophage differentiation [31,32]. Hypoxia further increases the presence of TAMs and TANs in the microenvironment, leading to increased vessel formation [5]. The number of CD4+ and CD8+ T cells has also been associated in increased vascular density [33]. Hypoxic tissue microenvironment also increases migration of natural killer (NK) cells and affects their chemotactic ability, thereby potentially affecting the formation of metastases [34].

After being disengaged from the tumor, circulating tumors cells (CTCs) can be killed by different immune cells, which can be prevented by forming multi-cellular aggregated with different immune cells and fibroblasts [35]. CTCs form aggregates with neutrophils, which causes changes in the expression profile of CTCs, favoring metastasis formation [36]. In pancreatic cancer, CTCs and neutrophils are found together and the presence of neutrophils predicts metastasis-free survival in patients [37]. A high number of CTCs with low number of T cells but high number of Tregs decreases immune surveillance in the tumor and results in metastatic disease [38,39]. Also, CTCs have been suggested to induce apoptosis of T helper cells via FAS-ligand pathway, leading to tumor immune escape and induction of dormancy [40]. After surviving the circulation, the CTCs need to extravasate to their new tissue microenvironment. Neutrophils can release neutrophil extracellular traps (NETs) that can promote metastases when they are in contact with CTCs [41,42]. Platelets can also interact with cancer cells to acquire invasive mesenchymal-like phenotype either by direct or indirect interactions through TGFβ and Wnt-β-catenin pathways [43,44,45].

Tumor cells prepare the new metastatic microenvironment already before their arrival, creating so-called pre-metastatic niche. Three major factors crucial for the formation of the pre-metastatic niche are primary tumor -derived components, tumor-mobilized bone marrow-derived cells and local stromal microenvironment of the host [46]. The pre-metastasis niche can be re-modelled by cancer secreted pro- or anti-metastatic exosomes. For example, anti-metastatic exosomes from melanoma cause expansion of monocytes that recruited NK cells and macrophages to kill the tumor cells [47]. On the contrary, exosomes with tumor RNA -activating TLR-3 induce chemokine secretion in lungs, leading to neutrophil recruitment and formation of lung metastases [48]. In pancreatic cancer, tumor exosomes cause formation of fibrotic microenvironment in the liver, leading to recruitment of TAM and growth of liver metastases [49]. In the pre-metastatic niche, hematopoietic progenitor cells differentiate into MDSCs, causing immunosuppression in the niche that later enables the growth of metastases [50]. For example, low levels of type I interferon (IFN) and high levels of granulocyte colony-stimulating factor (G-CSF) modifies the pre-metastatic niche, resulting in a high number of neutrophils in lung metastases and higher lung tumor burden [51]. On the contrary, if the pro-inflammatory process is blocked in the pre-metastatic niche, the myeloid cells are not recruited to the site, which disables formation of metastases [5].

As described above, the formation of metastases is greatly influenced by many immune cells. For this reason, it is rational to think that the use of immunotherapies could be beneficial for the prevention of formation of immunosuppressive microenvironment that would later lead to tumor cell escape for the primary location, and to disable metastases from forming and growing. Immunotherapies are currently studied in different metastatic cancers as mono- and combination therapies [52]. Targeting of immune cells in metastasis can be TME-dependent and many highly immunosuppressive sites, such as bone marrow, may be less responsive to immunotherapies [53,54]. Although metastases have been vastly studied in general, studying immune cells and immunotherapies has been less intensive, especially in preclinical animal models, despite one of the earliest studies dating back to the 1970’s [55]. This has been influenced by the lack of appropriate preclinical models. For better modeling of species-specific interactions between human tumor and immune cells, human immune cell engrafted mouse models have been established. These models and their utility in metastasis research will be discussed in the next sections.

## 2. Human Immune Cell Engrafted Mouse Models

The expanding research done in immuno-oncology has created a need for more translational models that would allow the study of interactions between human tumor and human immune cells and importantly, to predictably assess efficacy and safety of immunotherapies [56]. Most preclinical cancer studies have been performed in immunodeficient mice, allowing the use of human cancer cells, but these models are not suitable for immunotherapy testing as these mice lack immune cells at varying degrees depending on the used strain.

Preclinical models used in immunotherapy assessments have long been limited to using either syngeneic models with injected mouse cancer cells or inducible carcinogenesis models. Human and mouse proteins vary in structure and function, and although cross-reactive antibodies or small molecule inhibitors are developed, intracellular signaling [57,58] and polarization of immune cells [59,60,61] may differ between mouse and human cells. For this reason, murine cancer models are not always suitable for evaluation of efficacy and safety of immunotherapies. Interactions between human immune cells and human tumors cells need to be studied for better understanding of the function of immunotherapies in humans. This is also important to minimize the risk of failure in clinical trials [56] and to allow the entrance of more effective and safer products for patient use. To overcome this translational challenge, so-called humanized mice with functional parts of human immune system have been established during the past decades [62,63].

Humanized mice are commercially available, but many laboratories also produce their own mice for research needs. Some experimental protocols are also published for creating these mice [64,65,66]. Briefly, the two major choices for creating these mice that reflect the performance of the model in immuno-oncology research are the immune cells used to engraft the mice and the mouse strain used as a host for engraftment, which are discussed in the following paragraphs.

Mouse models of reconstituted human immune system (HIS) are often referred as HIS-models that can be created by injecting intravenously either peripheral blood mononuclear cells (PBMCs) or hematopoietic stem cells (HSCs) into recipient hosts at young age. The choice of cells used for engraftment affects the immune cell population in the models. T cells are most abundant in both models, but HSC engrafted mice also have other leukocytes such as B cells and NK cells. Furthermore, depending on the mouse strain used for engraftment, other immune cells may also differentiate, which is discussed later in this section. PBMCs are typically isolated from adult donors and HSCs typically from cord blood, but also human liver and thymus can be used as a source for HSCs [67,68]. CD34+ engrafted mouse models are often used in immuno-oncology and in metastasis research that are described later in this review. Another option for cell-derived humanized models is to use fetal tissue pieces of liver and thymus implanted under the renal capsule and HSCs intravenously or into bone marrow [69,70]. This BLT-model (stands for Bone, Liver and Thymus) shows full reconstitution of human immune system in the periphery, and the mice have almost complete human immune system. Furthermore, T cells educated in donor-matched thymus implants and liver implants provide important cytokines for immune cell maturation, which results in improved antigen presentation and immune responses in the model. The BLT-model can be extended by a secondary transfer of bone marrow and thymus implants from BLT mice to super-immunodeficient mice to create so-called propagated BLT mice [71]. In this model, the quantity of immune cells is comparable to the “original” BLT mice and less fetal samples are needed, making this method an easier and less material consuming approach in generating BLT mice. However, in another model, such serial transplantation of immune cells from mouse to mouse was lost after a couple of transplantations, and caution should be paid that an adequate number of immune cells is present in the model before its use in immuno-oncology studies [72].

PBMC engraftment allows studies to commence fast because immune cells mature within one to two weeks. However, study length is often limited because of the early onset of graft-versus-host disease (GVHD), limiting the study length to 3–4 weeks [73]. Though PBMC models provide possibilities to study different aspects resembling clinical human GVHD, in the aspect of oncology models can be used in short-term immune activation studies but they cannot be used for example in long-lasting efficacy studies. However, recent studies show that the onset of GVHD can be diminished by using immunodeficient mice deficient in murine MHC class I and II proteins [74,75] but so far, this approach has not been widely used in immuno-oncology research. HSC engrafted models are more time consuming as the differentiation of human immune cells from the progenitors takes about 16 weeks from engraftment. Compared to PBMC models, HSC models have more stable engraftment and they are only seldom reported to have GVHD at the late time point of 24 weeks after engraftment of immune cells, giving a time window of about 8–10 weeks for a oncology study [76]. BLT mice can also develop GVHD, limiting their use in long-term studies [77]. In addition to GVHD, humanized models may develop other severe symptoms such as anemia [78], potentially affecting their long-term survival.

The recipient host for human immune cells needs to be immunodeficient. The used mouse strains are most commonly super immunodeficient hosts such as NOG, NSG or BRG mice [67,79,80]. Typically before inoculation of CD34+ HSCs, the mice are preconditioned, causing a mild myelosuppression either by low-dose irradiation [81] or chemical induction for example using busulfan [82] or c-kit antibody [67] that deletes almost all endogenous mouse HSCs and enhances the engraftment of human cells. The myelosuppression typically done at young age results in higher reconstitution of human immune cells. However, in some mouse strains such as NBSGW, high engraftment of human cells has been observed even without irradiation [83]. Also, mice engrafted with PBMCs are not preconditioned. Naturally, all the mouse strains have differences in engraftment and in maturation of different immune cells [84] but all of them seem to have enough human cells causing biological effects and therefore be suitable for immuno-oncology studies.

The first used host for the injection of CD34+ cells was the NOD.Scid mouse that showed good engraftment of human immune cells in peripheral blood, spleen and bone marrow [85]. In these mice, human T and B cells are mainly developed and appropriate cytokines are being produced for immune activation. This model was later enhanced by using NOD.Scid IL-2 deficient mice as a host for CD34+ cell injection [86], and it showed higher amounts of human immune cells that lasted for over 24 weeks, suggesting that hematopoiesis was occurring in the mice. Furthermore, the engrafted cells differentiated into lymphocytes, platelets, erythrocytes and dendritic cells. The engraftment and maturation of human HSCs is typically verified by flow cytometry analysis for human CD45+ lymphocytes, and more detailed analysis of for example CD3 (T cells), CD4 (T helper cells), CD8 (cytotoxic T cells), CD56 (NK cells), CD11b (myeloid cells) can be performed. In peripheral blood, CD45+ cells can be detected at 6 weeks post-engraftment at low amounts that increases to 25% after 8 to 16 weeks, and to 30% after 22 weeks [87]. At 22 weeks, almost 80% of CD45+ cells are CD3+ T cells, 10% CD20+ B cells and the remaining 10% myeloid and NK cells. However, these percentages are not universal and the amount of immune cells in the mice depends at least on the age and strain of the mice, the method used to cause myelosuppression, the type of donor material and route and number of injected cells. In these models, T cells home to thymus where they mature to CD4+ or CD8+ cells and then travel to other lymphoid organs [88]. Importantly in the context of oncology, the matured CD8+ cells can inhibit tumor growth when the mice are treated with an immunotherapeutic agent [88].

In these above-mentioned HSC engrafted models, mainly T and B cell mediated effects can be studied due to low number of myeloid-derived cells. However, the myeloid-derived cells have an important role in TME and in metastasis, and the novel humanized mouse models need to also support the development of these cells. One of the first models developed was the transgenic human interleukin 6 (IL-6) NOG mouse [89]. When these mice are engrafted with human CD34+ HSCs they also develop into CD33+ myeloid cells, CD33+CD14+ monocytes and CD163+ TAMs that can also secrete more immunosuppressive cytokines such as arginase-1 than in NOG mice engrafted with the same cells. Besides an increased number of myeloid cells, these mice have a slightly higher rate of CD45+ cells in the circulation, spleen and bone marrow compared to NOG mice. Additionally, transgenic mice for human IL-3 and granulocyte macrophage colony-stimulating factor (GM-CSF) engrafted with CD34+ HSCs support the development of human myeloid cells of various lineages [90]. Another cell type important for metastasis research is NK cells whose function can be studied in IL-15 transgenic mice that support the differentiation of NK cells [91].

Humanized mouse models are relatively new and as they are more used in research, we will gain better understanding of their translational power and limitations in oncology research. It is important to understand the limitations of the models as they affect study planning and interpretation of the results. Major limitations include donor-caused and donor-independent differences in the engraftment and differentiation of human immune cells, and in a larger content, the variability between different mouse strains used as hosts for human immune cells. Limitations related to immune responses observed in these mice may include the lack of certain human cytokines that cause immune activation, and incomplete HLA-matching that causes a non-specific immune reaction especially in tumors with low antigen presentation. Finally, GVHD is a major limitation especially in PBMC models, which limits the timelines for performing a study, as PBMC engrafted mice develop GVHD within 3–4 weeks. On the other hand, HSC engrafted mice are already over 16 weeks old when entering the study and their long-term survival should be noted.

When deciding a model for immuno-oncology research, the main criteria should be the human cells of interest and how they are presented in the humanized mouse models. In immuno-oncology studies, follow-up of tumor size should not be the only main outcome, and attention should be paid to the expression of immune markers in TME and immune cell infiltration. In the next section we will focus on the use of humanized models in metastasis research.

## 3. Human Immune Cell Engrafted Mouse Models used in Metastasis Research and the Effects of Immunotherapies on Metastases

Humanized mouse models have not yet been widely utilized in metastasis studies. Development of metastases in humanized mice has mostly been monitored only from subcutaneous and orthotopic cancer cell injection models. Generally, it is thought that tumor growth, especially in subcutaneous models, does not vary if the models are performed in immunodeficient or humanized models [88]. Still, the situation might change dramatically in the context tumor type [87] or when performing studies in metastasis models [92].

Gammelgaard and colleagues studied the growth and metastasis of human melanoma A375 cell line and two human triple-negative breast cancer (TNBC) cell lines, MDA-MB-231 and MDA-MB-468 in CD34+ HSC transplanted BRGS mice [88]. In their study, metastases were monitored after surgical removal of primary tumors reaching the maximum size. Subcutaneously inoculated A375 cancer cells developed metastases to liver and lungs, but not to brain, which is a common site of melanoma metastasis in humans. The melanoma primary tumors and metastases were poorly infiltrated by tumor-infiltrating lymphocytes (TILs), presenting an immunologically cold tumor type. The TNBC cells inoculated orthotopically into the mammary gland developed metastases to liver and to lungs occasionally. These tumors had a high number of TILs that were associated with slower tumor growth. The lung metastases had higher density of TILs compared to the primary tumors or liver metastases of the same mice. Interestingly, the mice that did not develop any metastases had high density of TILs in the primary tumor.

Shafiee and colleagues developed an advanced bone metastasis model with both human bone (human tissue engineered bone, hTEB) and human CD34+ human immune cells [93]. When mice with subcutaneously transplanted TEB and engrafted with human CD34+ HSCs were 11 weeks later injected orthotopically with human MDA-MB-231BO metastatic TNBC cells, formation of primary tumor and metastases to hTEB was observed. However, the model with CD34+ cells resulted in lower number of metastases to hTEB and overall lower tumor burden to hTEB compared to a model with no prior injection of CD34+ cells. This indicates a role for immune cells against developing breast cancer bone metastases.

Wagner and colleagues also studied bone and used SAOS-2 human osteosarcoma cells in orthotopic humanized tissue engineered bone constructs (ohTEBC) placed in the femur of mice engrafted with human CD34+ cells [94]. Osteasarcoma caused new bone growth (an osteoblastic bone reaction) in the ohTEBC and the cells metastasized to lungs similarly to what is observed in osteosarcoma patients.

The above cited studies show that immune cells have a role in the formation and growth of metastases in preclinical models. The essential question that arises is: are different immunotherapies able to prevent or treat metastasis in these models? This has been evaluated, for example, in the models for TNBC, melanoma, bladder, adrenocortical carcinoma, osteosarcoma and bone metastasis (Table 1).

TNBC is unresponsive to hormonal therapies and often highly metastatic, which increases the need for new therapeutic strategies, including immune targeted therapies. Rosato and colleagues studied TNBC using the patient-derived xenograft (PDX) tumor line MC1 in CD34+ HSC-engrafted NSG mice in comparison to non-humanized NSG mice [87]. They observed that one main characteristics of the TNBC PDX model, capability to metastasize to lungs, remained in the humanized mouse PDX model and high number of TILs were observed in lung metastases. Anti-PD-1 therapy (nivolumab) reduced tumor growth and improved survival in humanized NSG mice. On the contrary, the use of another immunotherapy, anti-CTLA-4 (ipilimumab) as monotherapy did not show anti-tumor effects in the same model. Sai and colleagues studied a PI3K inhibitor (BMK120) in an orthotopically implanted TNBC PDX model in mice engrafted with CD34+ HSCs and followed by autologous infusion of CD4+ and CD8+ T cells of the same patient [95]. Inhibition of the PI3K signaling pathway by BKM120 inhibited tumor growth and lung metastasis by enhancing anti-tumor immunity by decreasing the number of CD4+ TILs, increasing the number of activated CD4+CD69+ T cells, and decreasing the number of CD20+ B cells. The number of CD8+ cytotoxic T cells that were scarce in tumors of humanized mice were unaffected by the treatment. The decreased number of lung metastases was suggested to be due to interrupted extravasation of cancer cells caused by enhanced anti-tumor immunity.

Metastatic melanoma has a poor prognosis and the development of novel therapies is an unmet clinical need [96]. Kuryk and colleagues studied the effects of an anti-PD-1 antibody (pembrolizumab) and an oncolytic virus with human GM-CSF (ONCOS-102) in melanoma models [97]. Human melanoma cell line A2058 was implanted into CD34+ huNOG mice, subcutaneously forming primary tumor and liver metastases. ONCOS-102 treatment as monotherapy or in combination with anti-PD-1 resulted in primary tumor volume reduction. However, liver metastases were not significantly affected by the treatments. Ferrari de Andrade and colleagues used a human A2058 melanoma cell line in an intravenous model in NSG mice reconstituted with human NK cells whose survival was supported by continuous injections of IL-2 [98]. Treatment with 7C6-hIgG1, a MHC class I chain-related protein A (MICA) α3 domain antibody decreased the number of lung metastases and the spread of metastases to liver. The decreased metastatic burden and better function of the liver enabled 7C6-hIgG1 to increase survival. Jespersen and colleagues studied melanoma in an autologous PDX model where tumor cells and TILs from the same patient were implanted into NOG/NSG mice and T cell survival was maintained with continuous presence of IL-2 [99]. After surgical removal of a subcutaneous PDX the tumors relapsed, and metastases formed in non-humanized hIL2-NOG mice. When these mice received autologous TILs from a patient that has exhibited an objective response to adoptive cell transfer in the clinic, the metastases diminished in almost all mice, showing that TILs can eradicate metastatic tumor growth.

In order to improve models where immunotherapies could be tested against bladder cancer and metastasis, Blinova and colleagues established subcutaneous PDX models of basal, luminal and p53 subtypes of primary and relapsed non-muscle invasive bladder cancer (NMIBC) in human lymphocyte transplanted mice [100]. All models that expressed PD-L1 developed lung metastases from primary tumors and the formation of metastases in all molecular subtypes of human primary and relapsed NMIBCs could be inhibited by treatment with the anti-PD-L1 antibody durvalumab.

Adrenocortical carcinoma (ACC) is an aggressive malignancy with very limited treatment options that eventually develops into metastatic disease in most patients. There is rationale to assess efficacy of immunotherapies in ACC [101], although modest clinical responses have been obtained so far. Lang and colleagues established a subcutaneous ACC PDX model obtained from a patient’s liver metastasis implanted in humanized CD34+ BRGS mice [102]. They evaluated efficacy of pembrolizumab and compared the response in a matched patient with progressive metastatic disease. In the PDX model, 60% inhibition of tumor growth and an increase in the number of CD8+ TILs and activated Granzyme B+ cells were observed with pembrolizumab. In the matched patient, a partial response with 79–100% reduction in the size of target lesions (in lung and liver) was observed with pembrolizumab, and with no new sites of metastasis. Pretreatment analysis of the patient’s liver metastasis also indicated a high number of CD8+ cells. Although this study demonstrated a similar response between a PDX model in humanized mice and a matched patient, the authors also concluded some limitations of the study, including differences in tumor growth rate and infiltration of TILs at different time points.

Immunotherapies have often been studied in context to solid tumors, and there are less studies relating to musculoskeletal tumors such as osteosarcoma. Zheng and colleagues studied the effects of an anti-PD-1 antibody (nivolumab) on osteosarcoma growth and metastases using human KHOS cells injected subcutaneously in a PBMC humanized mouse model [103]. Nivolumab had no effect on the subcutaneous tumor growth. However, less lung metastases were observed in nivolumab treated mice, indicating that the treatment suppressed metastasis formation. Corresponding with this observation, higher number of CD4+ and CD8+ cells were observed in lung metastases of nivolumab-treated mice, whereas no differences were observed in the number of these cells in primary subcutaneous tumors, and in immunohistochemical analysis of PD-L1 and PD-1 in tumors and lung metastases.

In context to bone, novel bone metastasis models have been established in humanized mice. While observing anti-tumor efficacy of pembrolizumab in a primary orthotopic TNBC model with MDA-MB-231(SA)-luc cells, no efficacy was observed against the tumor or tumor-induced bone changes in an intratibial model that mimics growth of bone metastases [104]. In a prostate cancer model with intratibially inoculated LNCaP cells in CD34+ humanized mice, no efficacy was observed with pembrolizumab against bone metastases [105]. It may be speculated that a tumor growing in the bone microenvironment may be less responsive to immunotherapies than a primary tumor with different TME. The low response rate of immunotherapies against bone metastases may be due to an observed very low number of CD8+ TILs and high number of CD4+ cells in the tumors growing in bone. The distinct properties of the TME in bone metastatic sites has also been observed in a clinical study with metastatic castration-resistant prostate cancer (mCRPC) patients treated with the anti-CTLA-4 antibody ipilimumab [106], demonstrating enhanced Th1 signature in soft tissue prostate cancer metastases. On the contrary, paired bone marrow samples did not have Th1 cells but instead had a significant increase in Th17 cells. The authors concluded this as mechanistic basis for the resistance of mCRPC to immunotherapy. Corresponding microenvironment-dependent differences in the function and differentiation for other types of immune cells may also exist and should be explored.

Taken together, the data available indicates that metastasis can be modelled in humanized mice and the metastatic prolife and immunological contexture of human tumors is preserved when the tumors are grown in humanized mice. Furthermore, both in preclinical models and patients, metastases can respond to immunotherapies differently than primary tumors. Also, metastases growing in different locations such as lung and liver may have differences in their response to immunotherapies. One promising example of translational power of humanized mouse models in metastasis research has been shown in ACC where preclinical findings of anti-PD-1 efficacy in humanized mice correlates with clinical responses. A more detailed analysis of metastases can help to guide immunotherapy development towards more effective novel compounds that could be beneficial especially for metastatic patients.

## 4. Immunotherapies and Adverse Events in Humanized Mouse Models

Immunotherapies induce immune-related adverse events (irAEs) that can occur in patients early or late after treatment [107]. As many irAEs are non-classical to chemo- or targeted therapies, it is more difficult to predict and monitor them in preclinical models. They can be systemic or organ-specific and may be different depending on the given immunotherapy, for example anti-CTLA-4 or anti-PD-1/PD-L1 monotherapy or treatment combinations [107]. These irAEs include typically pneumonitis, hepatitis, colitis, nephritis, dermatitis, encephalitis, and adrenal or pituitary insufficiency [108]. Importantly, unexpected irAEs have been observed in skeleton, causing compression, resorptive bone lesions and fractures [109], and inflammatory arthritis after the immunotherapy has been stopped [110]. Often these irAEs observed in patients cannot be predicted in traditional preclinical models because they are caused only by interactions with human cells [111]. Even many in vitro or in vivo models using non-human primates fail to predict these adverse effects due to species-related differences, complexity of immune system and different mechanisms inducing disease pathology. Therefore, model species that have human immune components may improve the ability to identify these adverse events and enhance safety of immunotherapies [112]. Some humanized mouse models have been used to identify adverse effects of immunotherapies, which are summarized in Table 2.

Weaver and colleagues studied nivolumab in a BLT humanized mouse model in NOG and transgenic hGM-CSF and IL-3 NOG (NOG-EXL) mice [108]. Though the humanized BLT-NOG mice had significantly reduced survival compared to humanized BLT-NOG-EXL mice; both strains showed similar adverse effects by nivolumab that are also observed in humans, including pneumonitis, hepatitis, nephritis, dermatitis and adrenalitis. Additionally, histological findings included pancreatic atrophy, myositis, and osteomyelitis in some mice.

The life-threatening complication cytokine release syndrome (CRS) is a potential adverse effect of immunotherapies [111] and testing has been difficult due to the lack of preclinical models. Yan and colleagues utilized a humanized BLT-model in multiple mouse strains (NSG, NRG, NOG) in order to establish an in vivo model for CRS. The immunosuppressant muromonab, a murine antibody against human CD3 receptor that induces CRS in patients, dose-dependently increased toxic effects and induced morbidity in the established models [112]. Muromonab increased levels of human TNF-α, IFN-γ, IL-2, IL-6, IL-8, IL-10, IL-13, IL-17A, IL12/23p40 and GM-CSF, and caused significant T-cell activation. Importantly, pretreatment with the glucocorticoid methylprednisolone delayed muromonab response for most of the cytokines. The pretreatment also decreased the T cell activation markers CD69, CD25 and CD45RO in spleen and bone marrow. However, it should be noted that glucocorticoids are themselves immunosuppressive agents and the study did not include methylprednisolone as monotherapy. Later Yan and colleagues studied the effects of a TGN1412 analogue, an anti-CD28 antibody that caused CRS and increased leukocyte number and activation markers [113]. In their study, the BLT model showed similar adverse effects that were observed in phase I clinical trials of the TGN1412 analogue, thereby demonstrating the translational power of this model in predicting irAEs.

The anti-CTLA-4 antibody ipilimumab that has been approved for the treatment of metastatic melanoma and advanced renal cell carcinoma is associated with severe irAEs, including rash and colitis [114]. Du and colleagues studied ipilimumab induced adverse effects in mice with humanized CTLA4 gene that recapitulated clinically observed cardinal features in mice [115]. In the model, ipilimumab, especially when combined with anti-PD-1 treatment, caused systemic T cell activation, and changed the ratio between regulatory and effector T cells. Importantly, anti-tumor effects without adverse effects were achieved in CTLA-4 heterozygous mice, while both anti-tumor and adverse effects were observed only in homozygous mice, indicating distinct genetic requirement for the adverse effects and anti-tumor efficacy. Furthermore, the bi-allelic engagement of CTLA-4 was a requirement for T cell conversion to regulatory T cells that caused the adverse effects in the model.

IL-2 based therapies have been accepted for the treatment of malignant melanoma and renal cell carcinoma. They expand T cells and are associated with dose- and dosing schedule-dependent severe toxicities. Li and colleagues developed a HIS model that recapitulated clinical symptoms of IL-2 immunotherapy, including vascular leak syndrome and cytokine storm [116]. The adverse effects of high-dose IL-2 were T cell mediated, caused by expansion of effector T cells and suppressive capacity of regulatory T cells.

Our own research also revealed unexpected adverse effects in a female CD34+ humanized mouse model [117]. Humanized mice supplemented with estradiol to support breast cancer tumor growth experienced severe anemia that was further worsened with concomitant treatment with anti-PD-1 (pembrolizumab) treatment, leading to decreased survival and early loss of mice. Anemia is also one of the most observed adverse effects in patients on anti-PD-1 treatment [118,119]. Importantly, anemia has been observed in humanized mouse models [78] and other immunocompetent models, and it has been shown to be cancer model dependent [120]. This can potentially affect the safety and efficacy endpoints observed in these models and precautions should be taken when interpreting the results.

Taken together, clinically observed irAEs can be predicted especially in BLT humanized mouse models. This is probably caused by more complete antigen presentation and enhanced immune activation in BLT models compared to HSC models as discussed earlier. Furthermore, these models provide an improved translational tool for studying adverse effects because they have the biological and functional human immune system. Species-related differences in predicting adverse effects has been shown for example in the TGN1412 analogue studies, where CRS was observed in humanized mice but not in other model systems such as non-human primates. As immunotherapies induce major irAEs in patients, these novel models should be increasingly used in early phases of preclinical development in order to develop safer products and predict the type and onset of irAEs in clinical trials.

## 5. Conclusions

Immunotherapies are an emerging treatment option for metastasizing cancers that are currently undergoing clinical investigations in many cancer indications. As discussed in this review, immune responses to metastases may vary depending on the tumor type and stage, taking into account the local tissue microenvironment, which makes it more challenging to interpret not only preclinical but also clinical findings. Lack of understanding interactions between immune system and metastasis in preclinical models highlights the need for further basic and translational research in this field. The data cited in this review shows the potential of humanized mouse models in metastasis research. However, new emerging data is still needed to show the true translational power of these models when preclinical findings are validated in clinical studies.

Novel preclinical metastasis models in humanized mice give us the possibility to understand the role of immune regulation in tumor growth and metastasis. Still, limitations of these models should be noted, and new preclinical metastasis models need to be established. Development of models with multiple humanized components and focusing especially on the metastatic tumor microenvironment will be of great value. These novel metastasis models will hopefully provide new opportunities for preclinical drug development both in terms of efficacy and safety assessment of immunotherapies.

## Figures and Tables

**Table 1 cancers-12-01615-t001:** Summary of anti-tumor effects of immunotherapies on metastasis in different humanized mouse models.

Cancer Type	Preclinical Oncology Model		Treatment	Anti-Tumor Effect on Metastasis	Ref
	Model Type	HIS Model			
TNBC	PDX model (sc)	CD34-NSG mice	Nivolumab and ipilimumab	Nivolumab anti-tumor effect, ipilimumab no effect	[87]
	PDX model (ot)	Autologous CD4 and CD8 T cells	BKM120	Decreased number of lung metastasis	[95]
	CDX model (ot and it)	CD34-NOG mice	Pembrolizumab	Decreased primary tumor growth but no effect in bone metastases	[104]
Melanoma	CDX model (sc)	CD34-NOG mice	Pembrolizumab and ONCOS-102	Anti-tumor effects in primary tumors, not effect on liver metastases	[97]
	CDX model (iv)	NK cell reconstituted NSG mice	MICA α3 antibody	Decreased number of lung and liver metastases	[98]
	PDX model (sc)	Autologous TILs in NOG and NSG mice	IL-2	Eradication of metastasis growth	[99]
Bladder cancer	PDX model (sc)	Lymphocyte transplanted mice	Durvalumab	Decreased growth of metastasis	[100]
ACC	PDX model (sc)	CD34-BRGS mice	Pembrolizumab	Tumor growth inhibition	[102]
Osteo-sarcoma	CDX model (sc)	PBMC model	Nivolumab	No effect on primary tumor growth but lowered the number of lung metastases	[103]
Prostate cancer	CDX model (it)	CD34-NOG mice	Pembrolizumab	No effect in bone metastases	[105]

Abbreviations: HIS = human immune system, TNBC = triple negative breast cancer, ACC = adrenocortical carcinoma, PDX = patient derived xenograft, CDX = cell line derived xenograft, sc = subcutaneous implantation, ot = orthotopic implantation, iv = intravenous implantation, it = intratibial (bone marrow) implantation, NK = natural killer, TIL = tumor infiltrating lymphocyte, PBMC = peripheral blood mononuclear cell, MICA = MHC class I chain-related protein A, IL-2 = interleukin 2.

**Table 2 cancers-12-01615-t002:** Summary of treatment-induced adverse effects in humanized mouse models.

Model	Treatment	Adverse Effects	Ref
BLT-NOG, BLT-NOG-EXL	Nivolumab	Pneumonitis, hepatitis, nephritis, dermatitis, adrenalitis	[108]
BLT-NSG, BLT-NRG BLT-NOG	Muromomab, TGN1412 analogue	Cytokine release syndrome	[112,113]
Transgenic CTLA-4 mice	Ipilimumab	Anemia, severe dilated cardiomyopathy, inflammation	[115]
CD34-BRGS	IL-2	Body weight loss, ruffled fur, loose stool, splenomegaly, nephrotoxicity, pulmonary edema	[116]
CD34-NOG	Estradiol supplement and pembrolizumab	Anemia, increased mortality	[117]

Abbreviations: BLT = bone, liver and thymus, CTLA-4 = cytotoxic T lymphocyte associated protein 4, IL-2 = interleukin 2.

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
