# Peer review of "Immunotherapies and Metastatic Cancers: Understanding Utility and Predictivity of Human Immune Cell Engrafted Mice in Preclinical Drug Development"

_cancers, 2020, doi:10.3390/cancers12061615_

Round 1

Reviewer 1 Report

In this review, the authors nicely summarized the current humanized mouse models in studying cancer metastasis, immunotherapies, and immune-related adverse events (irAEs). Since these models exhibit many limitations that some of them may be difficult to overcome, the interpretation of study results must be carefully addressed. Here are the reviewer’s suggestions:

  1. Line 129: “For this reason, murine cancer models are not always suitable for evaluation of efficacy and safety of immunotherapies.” It will be very helpful to provide a table to (1) compare features of immune system critical for cancer therapies between human and mice; (2) list the limitation of humanized mice (matching/unmatched HLA type, GVHD, individuality of the model, etc.).
  2. Line 162-166: here the authors discussed GVHD as one of the most significant limitations in humanized mice. Please discuss if only short-term/early-phase immune response can be studied in these models.
  3. The most critical part of disease modeling is translation. Therefore, please add “major discovery” and “clinical implication” in Table 1 and 2.
  4. For section 4, “Immunotherapies and adverse events in humanized mouse models”, have any study using the humanized models suggested the treatment to mitigate irAEs?  

Author Response

Response to Reviewer 1:

  1. Line 129: “For this reason, murine cancer models are not always suitable for evaluation of efficacy and safety of immunotherapies.” It will be very helpful to provide a table to (1) compare features of immune system critical for cancer therapies between human and mice; (2) list the limitation of humanized mice (matching/unmatched HLA type, GVHD, individuality of the model, etc.).

The journal limits the number of Tables in the review articles to two. We consider the two Tables that are currently in the review as the most important for the context of the work and would prefer to keep them. However, the question about differences between human and mouse immune system is indeed highly important especially for translational aspects. Even though this is of high importance, there are not yet summaries/review articles about this issue, and we believe that more detailed discussion about this would be a topic for another manuscript.

Related to the limitations of humanized mouse models, we modified the test (Lines 223-234) to summarize the limitations of the models.

  1. Line 162-166: here the authors discussed GVHD as one of the most significant limitations in humanized mice. Please discuss if only short-term/early-phase immune response can be studied in these models.

The text was modified to state that these models can be used in short-term studies (Lines 167-170).

  1. The most critical part of disease modeling is translation. Therefore, please add “major discovery” and “clinical implication” in Table 1 and 2.

Unfortunately, we could not find a way how to incorporate this information in the Tables without losing readability of the texts. Instead we added new text to the manuscript and the “major discovery and clinical implications” aspects are discussed in Lines 358-362 and Lines 433-438.

  1. For section 4, “Immunotherapies and adverse events in humanized mouse models”, have any study using the humanized models suggested the treatment to mitigate irAEs?  

To the best of our knowledge such studies have not yet been conducted. However, this is a very relevant and important question and we hope that it will be addressed in future studies.

Reviewer 2 Report

This is a timely review as humanized mouse models are becoming an important component of preclinical research in IO and beyond. 

Major comments:

Line 144: The statement is not entirely correct, as typically only activated T cells are present in PBMC-reconstituted immunocompromised mice; whereas, HSC-HIS mice have a more representative leukocyte subpoulation, including human CD45+ T, B and NK cell, as well as monocytes (albeit at low levels).

The authors acknowledge this in Line 191.

Line 195: The reported percentages are not universal and very much depend on a range of variable factors such as the humanization technique, age of mice at reconstitution, route of HSC injections, host, donor HSCs etc. These should be discussed.

Line 200: Authors should discuss the common caveat with commonly used HIS models of human cancers, whereby engrafted tumor cells and human leukocytes are not HLA-matched and hence T cell responses may not be tumor antigen specific per se. This is particulalry important in the case of cancers where the tumor neoantigen burden is low or the TME lacks TILs.

Other advances in the HIS mouse field have tried to overcome this, either by HLA typing donor and tumors or alternatively engineering HSCs to induce de novo cancers in HIS mice (eg, Kaur et al, J Immunol, 201; PMID: 30710044).

Minor comments:

Line 39: Revise sentence

Line 70: It is not clear to me how hypoxia can 'favor M1 macrophages differentiation'- this seems to be contradictory to its immunosuppresive nature?

Line 145: correct spelling for mouse strain

Line 227: Change 'chapter' to 'section'

Lastly, authors should consider citing recent articles on the utilization of HIS mice to predict adverse effects such as CRS (eg, TGN1412); and a relevant recent study using HIS models of metastatic breast cancer (Roghanian et al, Cancer Immunol Res, 2020; PMID: 31451483)

Author Response

Response to Reviewer 2:

  1. Line 144: The statement is not entirely correct, as typically only activated T cells are present in PBMC-reconstituted immunocompromised mice; whereas, HSC-HIS mice have a more representative leukocyte subpoulation, including human CD45+ T, B and NK cell, as well as monocytes (albeit at low levels). The authors acknowledge this in Line 191.

The sentence was modified to address this issue (Lines 145-148).

  1. Line 195: The reported percentages are not universal and very much depend on a range of variable factors such as the humanization technique, age of mice at reconstitution, route of HSC injections, host, donor HSCs etc. These should be discussed.

The text in Lines 203-206 was added to clarify this issue.

  1. Line 200: Authors should discuss the common caveat with commonly used HIS models of human cancers, whereby engrafted tumor cells and human leukocytes are not HLA-matched and hence T cell responses may not be tumor antigen specific per se. This is particulalry important in the case of cancers where the tumor neoantigen burden is low or the TME lacks TILs.

This issue is now shortly discussed in Lines 228-231.

  1. Other advances in the HIS mouse field have tried to overcome this, either by HLA typing donor and tumors or alternatively engineering HSCs to induce de novo cancers in HIS mice (eg, Kaur et al, J Immunol, 201; PMID: 30710044).

The de novo models in humanized mice are very interesting and relevant approach. In this review, we have limited the cited references only to those in which metastases from solid tumors in humanized mice are observed and that is why this paper was not cited in this work.

  1. Line 39: Revise sentence

The sentence was revised (Lines 37-40).

Line 70: It is not clear to me how hypoxia can 'favor M1 macrophages differentiation'- this seems to be contradictory to its immunosuppresive nature?

Thank you for the comment and we apologize for the inconvenience. There has been an error in the text and hypoxia indeed decreases M1 macrophage differentiation. This has now been corrected to the text in Line 70-71.

  1. Line 145: correct spelling for mouse strain

The mistake in the spelling was corrected (Line 147)

  1. Line 227: Change 'chapter' to 'section'

The suggested change was made (Lines 118 and 238).

  1. Lastly, authors should consider citing recent articles on the utilization of HIS mice to predict adverse effects such as CRS (eg, TGN1412); and a relevant recent study using HIS models of metastatic breast cancer (Roghanian et al, Cancer Immunol Res, 2020; PMID: 31451483)

Thank you for sharing the interesting articles. The paper by Yan et al., 2019 about CRS in humanized mice is now added to the manuscript (reference 115, Lines 403-407). In this review we have focused the cited work to those that report the use of immunotherapies in humanized mice. Therefore, the paper by Roghanian et al., 2019, was not added as they used chemotherapy in combination of targeted therapies (ERGF and HER2 targeted).

Reviewer 3 Report

This is a well written comprehensive and detailed review.

The PDX models are well described and the cited literature is very relevant.

An issue which is not mentioned is the interaction between the tumor cells and the mouse endothelial cells, an interaction which is relevant to the process of metastasis.

Author Response

Response to Reviewer 3:

  1. An issue which is not mentioned is the interaction between the tumor cells and the mouse endothelial cells, an interaction which is relevant to the process of metastasis.

This is a highly important interaction in regard to the metastatic process. However, the focus in this review is the interactions between human tumor cells and human immune cells and how these interactions relate to metastasis, and for this reason we have limited the discussion of interactions between other cell types to minimum.

Round 2

Reviewer 1 Report

The most important step in modeling is validation. Although most of the review comments were rejected by the authors, they need to at least explain how the modeling results, and thus their human relevance. can be validated.

Author Response

We thank the reviewer for the comment. The validation of these models can be finally achieved by comparing preclinical data to clinical findings. We have now included discussion about the translational aspects to our manuscript to highlight this issue. We have included discussion in lines 358-367, 441-445, and in the conclusions section, lines 456-459. All changes are highlighted in yellow.